# Use of Hugo^TM^ RAS in General Surgery: The First 70 Cases at a German Centre and a Systematic Review of the Literature

**DOI:** 10.3390/jcm13133678

**Published:** 2024-06-24

**Authors:** Orlin Belyaev, Tim Fahlbusch, Illya Slobodkin, Waldemar Uhl

**Affiliations:** Department of General and Visceral Surgery, St. Josef-Hospital, Ruhr-University Bochum, Gudrunstr. 56, 44791 Bochum, Germany; t.fahlbusch@klinikum-bochum.de (T.F.); i.slobodkin@klinikum-bochum.de (I.S.); w.uhl@klinikum-bochum.de (W.U.)

**Keywords:** robot-assisted, Hugo, general surgery, setup

## Abstract

**Introduction:** The versatile open modular design of the newly introduced robotic platform Hugo^TM^ RAS is expected to allow its rapid spread in general surgery. However, the system is not yet approved for use in oesophageal and HPB-surgery and is not licensed worldwide. The aim of this work was to review the current spectrum of general surgical procedures that may be feasibly and safely performed with Hugo. **Methods:** We retrospectively reviewed our own series and performed a systematic review of all the published reports of general surgical procedures performed with this system in the literature. **Results:** Seventy patients underwent general surgery with Hugo at our institution, and another 99 patients were reported in the literature. The most common procedures were colorectal (n = 55); cholecystectomy (n = 44); repair of groin, ventral and hiatal hernias (n = 34); upper GI (n = 28); adrenalectomy (n = 6); and spleen cyst deroofing (n = 2). No device-related complications were reported. Arm collisions and technical problems were rare. The docking and console times improved in all series. The port positions and robotic arm configurations varied among authors and depended on the surgical indication, patient characteristics and surgeon’s preference. **Conclusions:** A wide spectrum of general surgical procedures has been safely and effectively performed with the Hugo RAS, even by robotically inexperienced teams with a limited choice of instruments. Technical improvements to the system and the introduction of robotic energy devices may help Hugo evolve to a vital alternative to established robotic systems.

## 1. Introduction

The Hugo™ robotic-assisted surgery (RAS) (Medtronic, Dublin, Ireland) platform is one of several much-awaited alternatives to the DaVinci robot. It consists of a system tower; an open console, including a widescreen HD-3D display with dedicated glasses, two pistol-like handgrips as arm-controllers and a footswitch panel; and four individual arm carts with a wide manoeuvre range. The platform was first introduced in South America and the Asia–Pacific region and later became commercially available in other parts of the world. In the USA, the platform is still an investigational device that is not for sale and awaits FDA approval. Reports on the clinical use of Hugo first began appearing after the device had received its CE mark in the EU for use in gynaecologic and urologic surgery in October 2021 and for general surgery a year later. So far, about 60 Hugo™ RAS devices have been installed in Europe alone, showing promising results for prostatectomy, nephrectomy and hysterectomy [1,2,3]. It is believed that the open modular design of the platform would also allow its rapid adoption in general surgery. However, due to its novelty, the limited choice of instruments, setup guides that are under development and its missing certification in some parts of the world, published experience with Hugo-assisted general surgical procedures has been very limited. 

The aim of this study was to summarize the current experience with Hugo RAS in general surgery with a focus on the spectrum of indications, feasibility and safety of the procedures and technical features of the system through a systematic review of the existing literature along with our own early insights from the first 70 cases with this platform. 

## 2. Materials and Methods

### 2.1. Own Patients

A retrospective chart review was performed of all patients who underwent robotic-assisted general surgical procedures at our hospital after beginning work with the Hugo RAS in February 2023 until 31 May 2024. Due to regulatory issues in Germany, no procedures were performed between April and October 2023, so that the study period of active use of Hugo was 9 months. No specific exclusion criteria were applied for patient selection. All patients gave written informed consent for robotic surgery and were personally informed by the operating surgeon about the procedure in all its aspects, including its novelty and the limited experience with that system. They signed a special informed consent form including additional remarks about possible device failures and device-induced complications. This observational study was approved by the Ethics Committee of Ruhr University Bochum (No. 23-7872-BR). It was conducted in accordance with the Declaration of Helsinki. In addition to clinical perioperative parameters, exact data on trocar positioning, docking and console times and the technical performance of the device were gathered. All surgeries were recorded by the DS1 system of the platform and were reviewed via the Touch Surgery^TM^ 7.36 (Digital Surgery Ltd., London, UK) application. The docking time was defined as the time needed to attach all the manipulator arms to the instruments and to test them after trocar placement. The console time was the time between docking and final undocking. The total operative time was defined as the time from skin incision until skin closure. 

Our surgical team had previous extensive experience with open and laparoscopic surgeries but no clinical experience with robotic surgery. Extensive theoretical and practical training offered by the vendor was completed as already mentioned in a previous report by us [4]. The system components of the Hugo RAS platform have been repeatedly described in detail elsewhere [5,6,7].

### 2.2. Systematic Review of the Literature

As no MeSH terms for Hugo RAS have been established, two of the authors (OB and TF) performed an independent initial search in PubMed, Web of Science and Google Scholar with the exact text phrases “Hugo RAS”, “Hugo Medtronic” and “Hugo robotic surgery” to identify relevant records. All reports on Hugo RAS published until 31 May 2024, were screened, including an additional manual check of their reference lists, and those reports describing the clinical use of the platform in general surgery were retrieved in full-text and included in the summary. Publications in the fields of gynaecologic and urologic surgery, as well as those reporting duplicate or no patient results were excluded. Final records to be included were checked by all four authors of the review—disagreements and missing data were resolved by personally contacting the authors of the original studies and analysing the raw data when necessary. As all included studies were of the lowest evidence level, being retrospective case reports and low-volume series of strongly selected cases, no formal bias risk assessment was performed. Selection, attrition, performance, detection and reporting bias were characteristic for all selected records. The PRISMA flow diagram of the systematic review is presented in Figure 1. This systematic review is registered in PROSPERO (No. CRD42024558581).

## 3. Results

### 3.1. Own Patients and Procedures

The first two robotic-assisted general surgical procedures with the Hugo™ RAS at our centre included a cholecystectomy and a sigmoidectomy, performed on 13 February 2023. They, at the same time, represented the first-in-human procedures with the Hugo™ RAS in Germany. Since then, a variety of general surgical procedures have been performed in 70 patients. A summary of our initial experience with the Hugo RAS is presented in Table 1. Along with the listed 31 colorectal surgeries, a wide variety of concomitant procedures were performed, such as appendectomy, adhesiolysis, adnexectomy, fundoplication and rectopexy. There was no mortality in the group. All non-colorectal procedures were uneventful. Complications occurred only after major colorectal surgery: two presacral haematomas after rectosigmoid resections were treated via CT-guided drainage, and an anastomotic tear after an ultralow rectal resection was treated with endoluminal negative pressure. A high output stoma after another low anterior resection resulted in transient renal failure and had to be closed. An elderly patient received antibiotic treatment for a clostridial infection after a right hemicolectomy. Another one suffered a postoperative paralytic ileus, resulting in a prolonged hospital stay.

### 3.2. Patients and Procedures Reported in the Literature

The earliest report on the use of Hugo in abdominal non-gynaecologic and non-urologic surgery was published online by the Italian team of Raffaelli et al. in 2022, who described five transabdominal adrenalectomies [5]. In early 2023, the first three cases of partial colectomy were described by Bianchi et al. [6]. Currently, a total of 18 original papers by 11 different teams have been published, most of them coming from Italy (n = 7) and Spain (n = 4). A summary of these is presented in Table 2 [4,5,6,7,8,9,10,11,12,13,14,15,16,17,18,19,20,21].

### 3.3. Combined Data on Most Common Surgical Procedures

#### 3.3.1. Lower GI surgery

Following the first original case series of colectomies by Bianchi et al., several other authors reported the feasibility and safety of colorectal procedures with Hugo [6]. Romero-Marcos et al. published their experience including six rectal resections, three sigmoidectomies and a rectopexy [17]. Gangemi et al. reported a mixed series of patients who underwent various procedures, including five ileocecal resections, a right hemicolectomy and a sigmoidectomy [7]. Caputo reported recently on three rectal resections using different setups [16]. Our experience with Hugo includes 31 colorectal procedures, most of them being sigmoid resections (n = 11), rectal resections (n = 8) and right hemicolectomies (n = 8). A left hemicolectomy, an APR and two Hartmann’s reversals were also performed. A wide variety of setup configurations were used by different authors; however, similar operating times and clinical results were reported among all series. The butterfly 2 × 2 and the 3 × 1 configurations were the most commonly used, and the compact setup with the camera arm between the legs was preferred for low rectal resections (Figure 2).

#### 3.3.2. Upper GI surgery

The most common upper GI procedure was bariatric Roux-Y gastric bypass (n = 15), followed by a case series of 10 Heller myotomies for achalasia [12,15]. After publishing their initial results with RYGB in four patients, Raffaelli et al. reported later on a larger series of 15 morbidly obese patients treated with the use of Hugo [12]. Only the second report was included in this systematic review. Additionally, a sleeve resection and a subtotal gastrectomy were reported [7]. We performed a wedge gastric resection for a large benign tumour of the stomach. The setup for procedures in the upper abdomen is well established and widely accepted, making them feasible and safe (Figure 3).

#### 3.3.3. Cholecystectomy

Most publications on Hugo-cholecystectomy came from centres in Italy and Spain, reporting relatively small case series of up to seven patients. The largest reported single-centre study provided technical details and setup modifications for the procedure in 14 consecutive cases—it was published by our team in 2023 [4]. Meanwhile, we have performed 32 CCEs with excellent clinical results and no technical problems. A variety of trocar and robotic arm positionings for CCE were proposed by other authors [7,9,14].

#### 3.3.4. Hernia Repair

There are five reports solely focused on the repair of abdominal wall hernia with Hugo. Mintz et al. reported on the successful rTAPP of 13 groin hernia repairs in 10 patients [10]. Quezada et al. performed transversus abdominis release in 10 patients with large ventral hernias using redocking for each side [20]. The short shaft of Hugo’s instruments was pointed out by the authors as the main limitation. Jebakumar et al. described in detail the use of Hugo for two cases of rTAPP and five cases of ventral hernia repair in the IPOM technique [13]. Three robotic arms were used by all authors in cases of abdominal wall repair. Hiatal hernia repair as a separate procedure or as a part of fundoplication were described by Gangemi and Quijano in two cases [7,8]. We also performed three repairs of hiatal hernia: two of them with mesh reinforcement, and the third one combined with a Toupet fundoplication. In all cases of hiatal hernioplasty, four robotic arms in an upper abdominal setup were used. We do not perform the robotic repair of abdominal wall hernia because in Germany, this type of surgery is increasingly pushed into the strongly underfinanced outpatient care.

#### 3.3.5. Adrenals and Spleen

A case series of three left and two right adrenalectomies was published earlier by Raffaelli et al., using a three-arm setup and an assistant port in the periumbilical region [5]. We confirmed that a left adrenalectomy using a similar subcostal triangular positioning of the robotic trocars is safe and feasible; however, we placed the assistant trocar between the camera and right-hand trocars in the left lower abdomen. The Italian colleagues positioned all three arms at the same ipsilateral side of the patient, whereas we distributed two arms on the left side and one arm on the other side of the patient to avoid collisions and provide more freedom for the assistant. All cases were performed in a lateral decubitus position. The same setup but with four robotic arms and an assistant trocar was successfully applied by our team to perform the deroofing of large symptomatic spleen cysts in two very obese patients. 

#### 3.3.6. Technical Performance of Device

We experienced one technical problem with the device in the beginning of this series. In patient No. 3, one of the arms repeatedly did not recognize the instruments. The system had to be restarted twice until function was regained. The arm was successfully repaired before the next procedure. Also, a software update was necessary after the fifth procedure to eliminate some bugs. These technical problems led to a time delay but no adverse clinical events. Subtle arm collisions were repeatedly detected but tolerated by the device and did not interrupt the proper functioning of the system. Neither the instruments nor the other hardware parts broke or showed defects. There were no problems with the image or video transfer. All procedures are saved in the cloud and are easy to review at any time using the Touch Surgery^TM^ application on any device, allowing notifications, workflow analysis and video editing. There were no device-related patient injuries, neither in our collective nor in the literature.

## 4. Discussion

This paper offers a summary of all the available reported cases of general surgical procedures performed with the help of the Hugo RAS robotic platform so far.

As expected, since its introduction, the platform has mostly been used in the fields of urology and gynaecology. The noninferiority of Hugo RAS compared to DaVinci was already demonstrated in some large series of prostatectomies and nephrectomies [22,23]. The expectations of its successful use in general surgery are based on its open modular design allowing improved communication within the surgical team and the flexibility in the configuration of the four individual arms. The ergonomic position of the console for the surgeon, the pistol-like grips resembling laparoscopic ones and the lower price have been discussed as further possible advantages of Hugo [4,5,6,7]. The novelty of the device is inherently associated with some temporary disadvantages, such as the limited choice of instruments, lack of worldwide certification and approval for some major surgical procedures, immature training process, shortage of experienced proctors and difficulties in product supply.

The summary of our own initial experience and the review of the literature revealed that despite the abovementioned limitations, a wide spectrum of procedures in general surgery are feasible and safe to perform using the Hugo RAS. Interestingly, the clinical results and operating times of surgical teams without previous robotic experience, such as ours or that of Romero-Marcos, did not differ from those of robotically experienced authors. This paradox indicates that either Hugo RAS and DaVinci are quite different in terms of handling or that the learning curve with Hugo is steep and its interface extremely user-friendly. Moreover, our data provide evidence that general surgery with this system may be safe and feasible even in non-selected patient populations.

Indications for Hugo use showed geographical differences. Thus, reports on hernia repair came predominantly from countries outside Europe, just opposite to a series of colorectal resections and cholecystectomy. Such contrasts may reflect differences in health care reimbursement policies around the world. In Germany, current governmental attempts focus on redirecting all hernia repair into the underfinanced outpatient sector, which makes the use of robotic platforms for this indication unattractive. Another point is the strict hygiene regulation in the EU, which has rated the resterilisation of Hugo instruments insufficient and prohibited their multiple use so far. 

On an international level, Hugo was most often used for colorectal surgery. The specific challenges of these procedures include a wide multi-quadrant surgical field, extending from the mid/upper abdomen to the lower abdomen and pelvic region, and the use of multiple different instruments for the resection and reconstruction steps of the operation. The modular design of Hugo with four independent arms may offer an advantage to the single pivot point design of the DaVinci system; however, it lacks automatic targeting and procedural setup-memory. Thus, the manual setting of the docking and tilt angles is needed in every single patient, which may be challenging and requires profound device-specific experience of the surgical team. According to our experience and the available literature, the modular design of Hugo cannot fully prevent redocking in complex procedures such as low anterior rectal resections. The major limitations of the Hugo platform with respect to colorectal surgery include the lack of a wristed robotic advanced energy device and some other instruments, such as a clip-applicator and a linear stapler. ICG fluorescence is still not available for the Storz camera of Hugo. The shaft of Hugo’s instruments is significantly shorter than those of DaVinci, which is a problem in tall patients and those with a larger abdominal cavity, especially in multi-quadrant surgical procedures such as colorectal ones or the repair of large ventral hernias. A comparison of the features between the Hugo RAS and DaVinci platforms is summarized in Table 3.

The limited portfolio of robotic instruments currently available for Hugo requires more active participation from the bed-side assistant, who must apply clips, staplers and vessel sealing through an additional laparoscopic port. Increased interaction between the console and bed-side surgeon is therefore of utmost importance, and this is enabled by the open design of the platform. As the four robotic arms are relatively bulky and possess a large volume of movement outside the body, the assistant sometimes suffers limited manoeuvrability and must be careful not to be squeezed between the arms.

Some minor but annoying shortcomings of the Hugo system include the current time limit of 45 min for the use of the monopolar shears and the need to restart the whole system after every single procedure. Attention should be given to positioning the surgical table as high as possible to allow an adequate volume of movement for the arms with a positive tilt—this may sometimes lead to a higher risk of contact between the long sterile robotic arms and the non-sterile parts attached to the ceiling such as the lamps or anaesthesia devices.

The initial lack of clinically proven setup guides has led to a variety of setup configurations being used by different authors for the same procedure. It was especially obvious in colorectal surgery, where the number and positioning of ports, the positioning of the arms and the number of redockings differed substantially among the authors. However, these differences appeared to not affect the docking times, duration of surgery or clinical outcomes, providing evidence that the modular design of Hugo may effectively allow a personalized setup considering both the individual patient’s characteristics and the surgeon’s preferences.

Despite its comprehensive character and detailed presentation of the current Hugo standards in general surgery, this systematic review suffers some major limitations. The most important flaw comes from the low evidence level of the included studies and their heterogeneity: 18 studies reported on less than 100 patients treated for a variety of diseases and retrospectively described the outcomes of a relatively broad spectrum of procedures. Furthermore, the lack of established setup guides, the preferences and individual robotic experience of surgeons and the selection bias made the reported results hardly comparable. Another drawback is the lack of a direct comparison of Hugo to DaVinci as the current gold standard in robotic surgery and to other newly introduced robotic platforms, especially in terms of cost effectiveness. 

Adding our own experience with Hugo in 70 patients to the results of this systematic review aimed to show the feasibility and safety of this platform in general surgery and present a firsthand point of view of its advantages and limitations. However, this additional data cannot overcome the already mentioned drawbacks of the review regarding heterogeneity and the small patient numbers. 

In conclusion, the Hugo RAS platform allowed a wide spectrum of abdominal surgical procedures to be safely performed. Ongoing technical development and software updates of the system are expected to improve its performance. To ensure a more rapid dissemination and the increasing use of the system in general surgery, joint international efforts including the creation of a procedural register and the establishment of standard clinically proven setup guides have to be endorsed. 

## Figures and Tables

**Figure 1 jcm-13-03678-f001:**
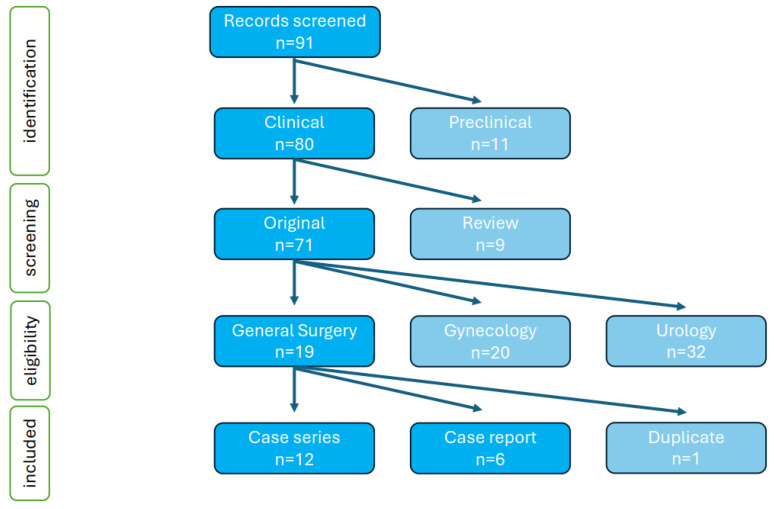
PRISMA diagram of the systematic review of the literature March 2022–May 2024.

**Figure 2 jcm-13-03678-f002:**
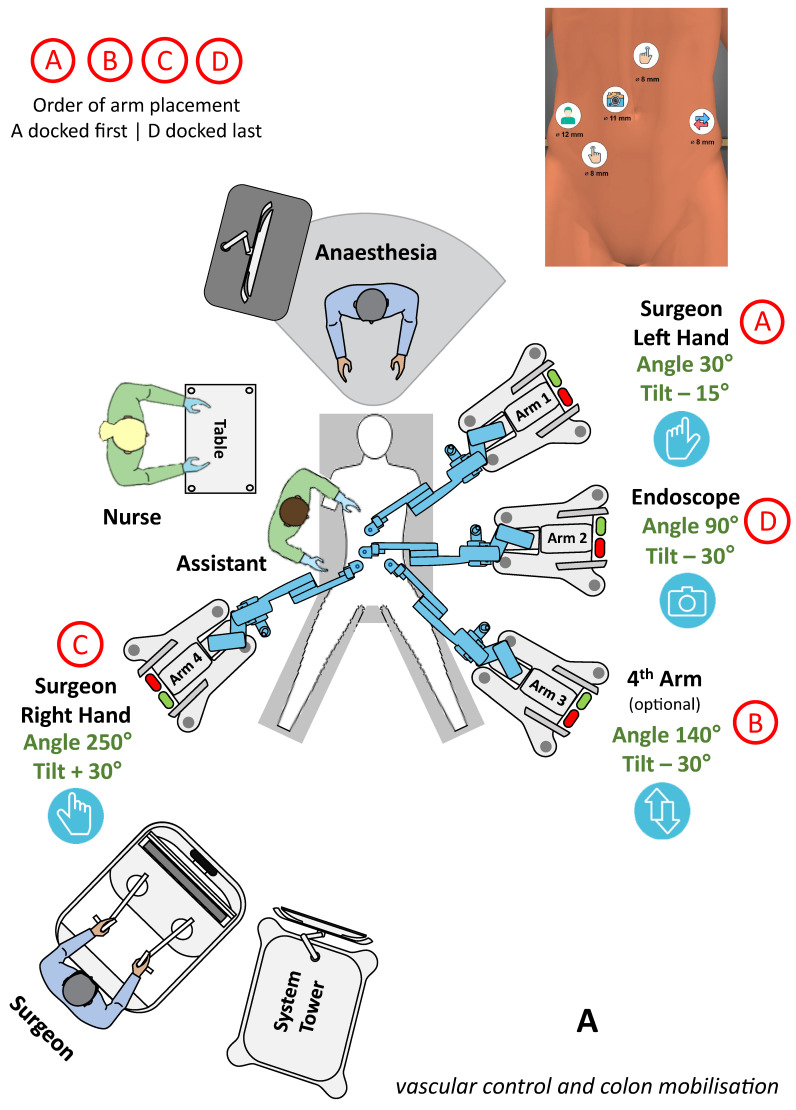
(**A**) Lower GI setup for vascular control and colon mobilisation. (**B**) Lower GI setup for pelvic mobilisation and anastomosis.

**Figure 3 jcm-13-03678-f003:**
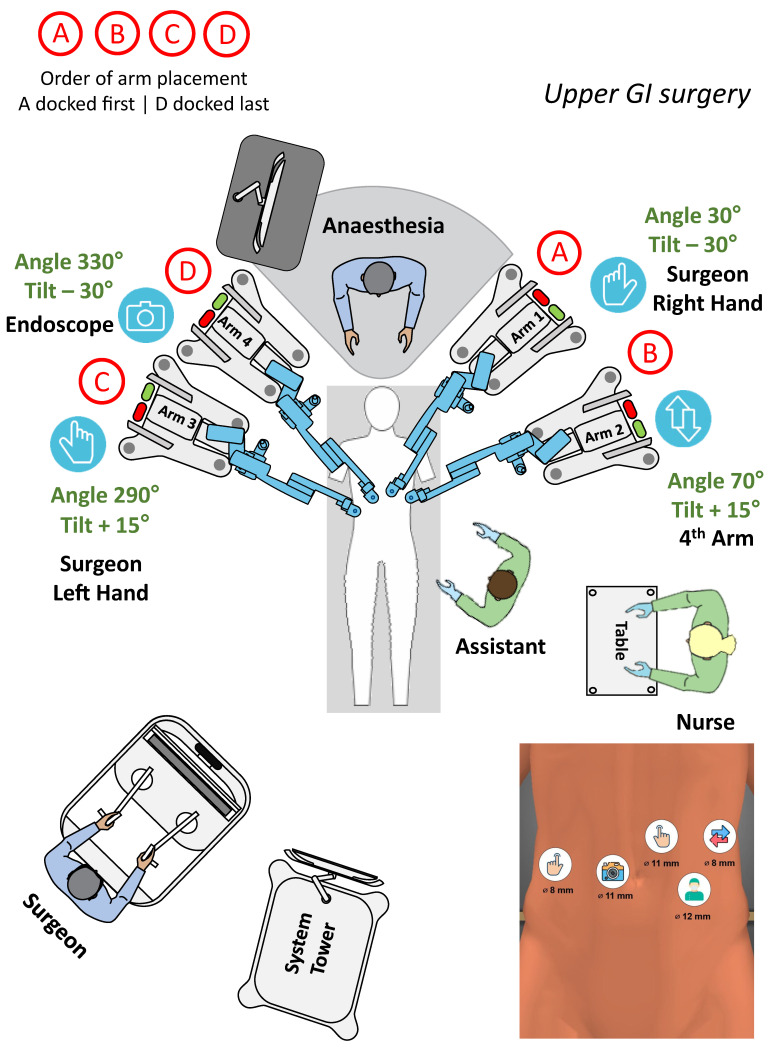
Setup for upper GI surgery.

**Table 1 jcm-13-03678-t001:** The first 70 general surgical procedures with Hugo^TM^ RAS at our centre over a period of 9 months.

Surgery	Diagnosis	Patients and Sex Distribution (F/M)	Age [Years]	BMI [kg/m^2^]	Dock Time [min]	Console Time [min]	OP Time [min]	Morbidity Major/Minor	Stay [Days]
Cholecystectomy	Lithiasis	32 (23 F/9 M)	62 (16–86)	28 (19–35)	10 (3–15)	45 (21–172)	72 (57–182)	0/0	2 (1–5)
Sigmoidectomy	Diverticula	11 (6 F/5 M)	66 (52–77)	25 (21–37)	10 (7–18)	120 (70–183)	189 (115–249)	0/1	7 (5–12)
Rectum resection	Cancer	8 (2 F/6 M)	66 (41–85)	27 (21–34)	12 (7–15)	233 (120–332)	325 (290–420)	2/2	10 (7–34)
Right colectomy	Cancer	8 (5 F/3 M)	67 (61–87)	27 (21–30)	12 (10–15)	180 (85–316)	219 (193–280)	1/1	7 (6–14)
Hiatoplasty	Hiatal hernia	3 (2 F/1 M)	47 (24–51)	27 (21–36)	10 (6–12)	151 (120–169)	197 (163–211)	0/0	6 (5–7)
Hartmann reversal	Colostomy	2 (1 F/1 M)	50 (33–67)	27 (24–30)	13 (12–14)	142 (120–164)	278 (264–290)	0/0	7 (7–7)
Cyst deroofing	Splenic cyst	2 (1 F/1 M)	55 (44–66)	38 (36–40)	15 (13–17)	90 (81–100)	130 (117–143)	0/0	3 (3–3)
Left colectomy	Cancer	1 (M)	68	23	12	155	228	0/0	6
Rectum exstirpation	Cancer	1 (F)	65	30	15	240	360	0/0	13
Gastric resection	Schwannoma	1 (M)	68	26	13	115	147	0/0	5
Adrenalectomy	Adenoma	1 (F)	69	19	12	150	200	0/0	6

For single cases, actual values are shown; for multiple cases, median values with (minimum–maximum) range are shown. F—female; M—male.

**Table 2 jcm-13-03678-t002:** A summary of published data on general surgery with Hugo^TM^ RAS in chronological order until 31 May 2024.

First Author	Centre	Year	Surgery	Diagnosis	Patients	Age [Years]	BMI [kg/m^2^]	Dock Time [min]	Console Time [min]	OP Time [min]	Stay [Days]
Raffaelli et al. [5]	Italy–Rome	2023	Adrenalectomy	Various	5	61	25	6	61	119	3
Bianchi et al. [6]	Italy–Milan	2023	Colectomy right, left	Cancer	3	72	26	8	260	350	6
Quijano et al. [8]	Spain–Madrid	2023	Nissen fundoplication	GERD	1	65	-	3	-	100	3
Gangemi et al. [7]	Italy–Bologna	2023	Colectomy various	Various	7	60	24	7	141	210	8
			Cholecystectomy	Lithiasis	7	51	26	6	50	97	1
			Gastrectomy partial	Various	2	53	26	5	140	186	8
			Nissen fundoplication	GERD	1	67	26	6	150	175	4
Belyaev et al. [4]	Germany–Bochum	2023	Cholecystectomy	Lithiasis	14	61	27	10	77	95	2
Vicente et al. [9]	Spain–Madrid	2023	Cholecystectomy	Lithiasis	1	54	-	3	-	70	2
Mintz et al. [10]	Israel–Jerusalem	2023	Hernia repair	Groin hernia	10	-	-	10	50	75	-
Caruso et al. [11]	Spain–Madrid	2023	Right colectomy	Cancer	1	-	20	5	-	200	8
Raffaelli et al. [12]	Italy–Rome	2024	Roux-Y gastric bypass	Obesity	15	48	42	7	100	150	2
Jebakumar et al. [13]	India–Chennai	2024	Hernia repair	Hernia various	7	-	-	23	-	128	2
Caputo et al. [14]	Italy–Rome	2024	Cholecystectomy	Lithiasis	3	59	26	10	50	98	2
Salem et al. [15]	Israel–Jerusalem	2024	Heller’s myotomy	Achalasia	10	43	23	10	-	130	2
Caputo et al. [16]	Italy–Rome	2024	Rectal resection	Cancer	3	71	25	12	338	415	8
Romero-Marcos et al. [17]	Spain–Barcelona	2024	Colorectal resections	Various	10	67	28	14	-	200	3
Wen et al. [18]	Taiwan–Changhua	2024	rTAPP	Groin hernia	1	33	-	-	-	-	1
Formisano et al. [19]	Italy–Milan	2024	rTAR	Incisional hernia	1	-	-	15	75	95	2
Quezada et al. [20]	Chile–Santiago	2024	Hernia rTAR	Ventral hernia	10	61	31	50	243	373	3
Toyota et al. [21]	Japan–Sapporo	2024	Rectum exstirpation	Rectal cancer	1	68	28	-	163	450	13

GERD—gastroesophageal reflux disease; rTAPP—robotic transabdominal patch plasty; rTAR—robotic transversus abdominis release.

**Table 3 jcm-13-03678-t003:** Comparison of the major features of the Hugo and DaVinci robotic platforms.

Features	Hugo RAS	DaVinci
Platform design	Modular with 4 independent arms	Single pivot tower
Ergonomics of surgeon console	Free head and neck movement	Constant head and neck strain
Team communication	Direct and easy through open design	Limited, no eye contact to rest of team
Hand controls	Laparoscopy-like pistol grips	Endo-wrist finger grips
Resolution and image quality	Excellent via dedicated 3D-glasses	Excellent via console binoculars
Side observers	3D-glasses allow multiple observers	Dual console allows a single 3D-observer
Teaching	Difficult due to single console	Easy due to dual console
Instruments	Short shaft, single use, limited choice	Great variety of instruments, multiple use
Setup configuration	Individual setup for every patient	Automatic setup after targeting
Docking memory	Not available	Yes, procedure specific
Vessel sealing device and linear stapler	Wristed and straight, not yet certified	Wristed, available
Camera	0° and 30°, 10 mm, no ICG	0° and 30°, 8 mm, ICG
Proctorship	Limited number of proctors	Plenty of experienced proctors
Training program	Short, basic, still in development	Established variety of courses
Role of bed-side assistant	Crucial, active	Supporting, passive
Video recording, editing and sharing	Via Touch Surgery app with AI-analysis	Via Intuitive Hub

## Data Availability

All raw data analysed in this article are available and saved on the data server of Katholisches Klinikum Bochum, and access may be requested at any time by addressing the Ethics Committee of Ruhr University Bochum.

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
