# Peer review of "Use of HugoTM RAS in General Surgery: The First 70 Cases at a German Centre and a Systematic Review of the Literature"

_jcm, 2024, doi:10.3390/jcm13133678_

Round 1
Reviewer 1 Report
Comments and Suggestions for Authors
Dear authors, thank you for the opportunity to review your manuscript. You brought up an interesting issue regarding the emergence of new robotic platforms.
Those are my considerations and suggestions regarding the study:
As you've conducted a systematic review, it's important to provide details about your search strategy and MeSH terms. This will enhance the reproducibility of your study. Additionally, a clearer delineation of your study selection and data extraction methods, along with an expansion of your search to more databases, will strengthen the comprehensiveness of your review.
Please add critical appraisal of study quality (biases risk assessment) to the included studies. I suggest the Joanna Briggs Institute (JBI) Manual for Evidence Synthesis.
Is there a registration number at the International Prospective Register of Systematic Reviews (PROSPERO)? Please add this information.
-In Table 1, please include minimum and maximum values for dock time, console time, operative time and length of stay.
-In the discussion section, it is imperative to delineate the drawbacks of the study.
Author Response
Point-to-point Reply to Reviewer 1:
Dear authors, thank you for the opportunity to review your manuscript. You brought up an interesting issue regarding the emergence of new robotic platforms.
Dear Reviewer, thank you for the fast review and the positive overall evaluation of the manuscript. We are grateful for your comments and did our best to introduce all recommended corrections and provide the additional information required in the text and tables. Please, find our detailed point-by-point reply attached here.
Those are my considerations and suggestions regarding the study:
As you've conducted a systematic review, it's important to provide details about your search strategy and MeSH terms. This will enhance the reproducibility of your study. Additionally, a clearer delineation of your study selection and data extraction methods, along with an expansion of your search to more databases, will strengthen the comprehensiveness of your review.
The novelty of the Hugo platform made the search of the literature somewhat easier. The first reports on Hugo were published in 2022 as the system was licensed for human use, so the search retrieved results for a short time of 2 years till 31.05.2024. Neither “Hugo” nor “Hugo RAS” are established MeSH terms, so we followed the recommendation of PubMed and searched for the exact text words “Hugo”, “Hugo RAS” and “Hugo robotic”, whereas the “Hugo RAS” gave the most comprehensive results in PubMed. We used the same words to screen Web of Science and Google Scholar for further results, however no additional reports were found. After completion of the initial screening, the reference lists of all publications were manually checked for other published material which may correspond to the inclusion criteria of the review. The results are presented in the PRISMA diagram of the review. All above mentioned databases were checked several times by the 4 authors of this manuscript and we can guarantee the comprehensiveness of the presented results. This detailed information of our search strategy is now added to the methods section of the manuscript (lines 86-102).
Please add critical appraisal of study quality (biases risk assessment) to the included studies. I suggest the Joanna Briggs Institute (JBI) Manual for Evidence Synthesis.
We acknowledge the importance of bias assessment as pointed out by the Reviewer and included now a comment on this issue in the methods section (lines 97-100). As all published reports on Hugo RAS in general surgery are case reports and small case series of lowest evidence level, no bias risk assessment can really be performed. Obviously, all included studies were strongly biased. A selection bias is present in all selected reports, since patients for Hugo surgery were carefully selected by the authors and this was clearly admitted in most of the publications. Performance, detection, and attrition biases were also present in most of the studies due to incompleteness of data and no blinding of surgical teams and patients. Reporting bias couldn’t be excluded with certainty due to missing information about non-reported cases (e.g. conversions to laparoscopic or open surgery due to hardware/software problems or intraoperative complications).
Is there a registration number at the International Prospective Register of Systematic Reviews (PROSPERO)? Please add this information.
Our review was submitted for publication in the PROSPERO Register and is currently under review (ID 558581) awaiting registration. As soon as the official registration number is assigned to the review it will be forwarded to the Editor of the journal. We expect this to happen next week. This information is added in the methods section (lines 101-102).
-In Table 1, please include minimum and maximum values for dock time, console time, operative time and length of stay.
We included all minimum and maximum values for the subgroups in Table 1, not only for the times and length of stay, but also for age and BMI, as well as details of the gender distribution for completeness of data. The completely revised Table 1 is now submitted with the revised manuscript. (lines 131-134).
-In the discussion section, it is imperative to delineate the drawbacks of the study.
We added a paragraph in the discussion pointing out all major and minor drawbacks of our study (lines 319-332). The most important flaw of our review comes from the low evidence level of the included studies and their heterogeneity: 18 studies reported on less than 100 patients, treated for a variety of diseases and retrospectively describing the outcomes of a broad spectrum of procedures. Furthermore, the lack of established setup guides, the preferences and individual robotic experience of surgeons and the selection bias made reported results hardly comparable. We tried to overcome some of these disadvantages by adding our own original unpublished data on 70 procedures to the systematic review - thus we aimed to give some own insights into the technical aspects of the Hugo use for general surgery and bring more objectivity when presenting the advantages and limitations of this robotic platform. Another drawback is the lack of direct comparison of Hugo to DaVinci as the current gold standard in robotic general surgery. However, such comparative studies will need several more years to come. Finally, no cost comparison to other robotic platforms is possible now.
Reviewer 2 Report
Comments and Suggestions for Authors
Dear authors, first of all I want to congratulate you for your work! It is a very interesting and well presented subject. The reader will definitely learn from this presentation.
It is very crucial that you presented the limitations of this robotic system and I will emphasize them, due to DaVinci world wide approval.
The Main question that must be answered in this article and from which your conclusions must be raised is the following; why someone use this device Ruther than the DaVinci? [ You have presented very clearly all the flaws about the assistant (he is more active ,although in this way the surgeon is more dependent. ), you said about the colorectal surgeries and instruments, etc.]
Also, I think that a table comparing the two systems will be helpful.
Finally, I want to point out that is very right that you explained by yourselves the limitations.
Author Response
Point-to-point Reply to Reviewer 2:
Dear authors, first of all I want to congratulate you for your work! It is a very interesting and well presented subject. The reader will definitely learn from this presentation.
Dear Reviewer, we are grateful for your prompt and very positive evalution of our manuscript. We did our best to follow your recommendations and included additional information in the manuscript, as required by you.
It is very crucial that you presented the limitations of this robotic system and I will emphasize them, due to DaVinci world wide approval.
Some of the limitations of the Hugo robotic platform were already pointed out in the discussion section. In order to give a comprehensive and clear comparison of Hugo and DaVinci we added a Table 3, summarizing the advantages and disadvantages of both platforms, as described by other authors and based on our own observations. Also, several other comments on the limitations of the system were added to the discussion (lines 288-297, 307-311).
The Main question that must be answered in this article and from which your conclusions must be raised is the following; why someone use this device Ruther than the DaVinci? [ You have presented very clearly all the flaws about the assistant (he is more active ,although in this way the surgeon is more dependent. ), you said about the colorectal surgeries and instruments, etc.]
We have to admit, this is a central question in the discussion about Hugo, but also about any other newcomer on the scene of robotic surgery (CMR, Versius, etc) – currently, the answer is not easy and not straightforward. Of course, DaVnci is still the gold standard due to its advantage of 25 years monopolistic development and this platform will continue to lead the market in the close future. However, Medtronic is financially strong and has a much wider market presence with a lot of other products and is an established manufacturer of high-quality surgical instruments. They can afford themselves the production, rapid development and maintenance of a high-end robotic platform at a lower price than Intuitive. Some features of Hugo such as its open design, the modular mobile setup and the freedom of configuration, as well as the emerging AI-analysis of surgical videos through the Touch Surgery App may give it some advantage when choosing a robotic surgical platform. Also, Hugo uses its own instruments, but a cooperation with Karl Storz as supplier for camera and instruments exists. Our own decision to buy a Hugo was based on a mixture of the above mentioned considerations mixed with some pioneering spirit. We also do not have urology and gynecology units at our hospital, so we are free to use the device only in visceral surgery. Overall, the development of DaVinci alternatives will results in more competitive environment and have a positive effect on the development and distribution of robotic surgery worldwide.
Also, I think that a table comparing the two systems will be helpful.
We find this a very good idea and a comparison Table 3 is now added to the manuscript. (lines 295-297)
Finally, I want to point out that is very right that you explained by yourselves the limitations.
Thank you for this comment – because of the small number of reports, we intentionally added unpublished original data based on our own experience. This allowed us to discuss the limitations of the system at first hand, which we hope may be of advantage to the audience.
Round 2
Reviewer 1 Report
Comments and Suggestions for Authors
No further comments.
Author Response
Thanks for the positive review!